# PCL/Collagen/UA Composite Biomedical Dressing with Ordered Microfiberous Structure Fabricated by a 3D Near-Field Electrospinning Process

**DOI:** 10.3390/polym15010223

**Published:** 2022-12-31

**Authors:** Zhirui Mai, Qilong Liu, Yongshuang Bian, Peng Wang, Xuewei Fu, Dongsong Lin, Nianzi Kong, Yuqing Huang, Zijun Zeng, Dingfan Li, Wenxu Zheng, Yuanjun Xia, Wuyi Zhou

**Affiliations:** 1Key Laboratory for Biobased Materials and Energy of Ministry of Education, Biomass 3D Printing Materials Research Center, College of Materials and Energy, South China Agricultural University, Guangzhou 510642, China; 2Department of Pharmaceutical Engineering, College of Materials and Energy, South China Agricultural University, Guangzhou 510642, China; 3Graduate School of Guangzhou University of Chinese Medicine, Guangzhou 510006, China; 4Guangdong Key Lab of Orthopedic Technology and Implant Materials, Department of Orthopaedics, General Hospital of Southern Theater Command of PLA, The First School of Clinical Medicine of Southern Medical University, Guangzhou 510010, China

**Keywords:** near-field electrospinning, collagen, antibacterial, biomedical dressing

## Abstract

In this work, a functionalized polycaprolactone (PCL) composite fiber combining calf-type I collagen (CO) and natural drug usnic acid (UA) was prepared, in which UA was used as an antibacterial agent. Through 3D near-field electrospinning, the mixed solution was prepared into PCL/CO/UA composite fibers (PCUCF), which has a well-defined perfect arrangement structure. The influence of electrospinning process parameters on fiber diameter was investigated, the optimal electrospinning parameters were determined, and the electric field simulation was conducted to verify the optimal parameters. The addition of 20% collagen made the composite fiber have good hydrophilicity and water absorption property. In the presence of PCUCF, 1% UA content significantly inhibited the growth rate of Gram-positive and negative bacteria in the plate culture. The AC-PCUCF (after crosslinking PCUCF) prepared by crosslinking collagen with genipin showed stronger mechanical properties, water absorption property, thermal stability, and drug release performance. Cell proliferation experiments showed that PCUCF and AC-PCUCF had no cytotoxicity and could promote cell proliferation and adhesion. The results show that PCL/CO/UA composite fiber has potential application prospects in biomedical dressing.

## 1. Introduction

In recent years, biological 3D printing technology has attracted much attention in biomedical fields such as regenerative medicine engineering, surgical simulation, and disease diagnosis, because it can customize functional organs and tissue structures according to individual characteristics of different patients, and has made some important research progress [1,2,3]. Furthermore, the electrospinning method can be used to conveniently and quickly prepare medical electrospun fiber materials containing biological materials, skin growth factors, and pharmaceutical ingredients for preparing porous biomedical materials [4,5,6]. The controlled distribution of drugs and other components in electrospinning fibers can be used for precise treatment and repair of skin wounds [7,8]. For the research focus in the field of electrospinning in recent years, coaxial electrospinning can be used to prepare shell-core fibers, but the fibers are often disordered [9,10]. As a new spinning method, melt electrospinning has the advantages of no solvent and environmental protection compared with traditional solution electrospinning [11]. However, it needs heating, and the available materials are mostly low melting point materials, which limits its development. Near field electrospinning (NFES) is a new fiber manufacturing technology that has emerged in recent years. It combines biological 3D printing and traditional disordered electrospinning technology to prepare highly ordered microfibers. Zhong and his colleagues prepared PCL fibers/GelMA composite materials with NFES, which has a good effect on promoting the healing of diabetes wounds, showing the broad application prospect of NFES in medical dressings [12]. Theoretically, the ideal wound dressing should have the advantages of hydrophilicity, antibacterial activity, water absorption property, controllable mechanical properties, biocompatibility, and air permeability [13,14].

Collagen, as the basic structural protein of extracellular matrix, has broad application prospects in the fields of food, food packaging, biomaterials, and cosmetics because of its unique biological activity [15,16]. Collagen has excellent hemostatic performance, low immunogenicity, biodegradability, and the performance of promoting cell proliferation and differentiation, making it the first choice of medical materials, which can be used for the preparation of new hemostatic materials, scaffold materials, artificial skin, artificial blood vessels, etc. [17,18]. However, as a kind of natural macromolecular organic matter, the traditional collagen membrane has many problems, such as poor mechanical properties, poor thermal stability, easy degradation in water, fast degradation time and so on, which limits its wider application in these fields [19]. Therefore, solving the defects of collagen macromolecules themselves and developing a new type of high-performance collagen composite fiber material, so that it can be more widely used in the field of biomaterials, are some of the hot spots in the research field today.

At present, biodegradable synthetic polymers widely used in tissue engineering scaffolds mainly include Polylactide (PLA), Polycaprolactone (PCL), Polyurethane (PU), and other polymer materials [20,21,22]. They have a controllable degradation rate, good operability, excellent mechanical properties, non-toxic side effects, and other advantages within a certain range, and have been extensively studied in the field of electrospinning. It was found that they can successfully construct micro and nano fiber scaffolds through electrospinning technology. Among them, polycaprolactone has excellent biocompatibility, permeability and is non-toxic, and has a very wide range of uses in medicine [23]. PCL/PLA nanofiber-covered yarns prepared by Herrero and his colleagues have a good effect on promoting cell proliferation [24]. The PCL/fibrin vascular stent prepared by Zhao et al. is a small diameter tissue engineered vascular stent with good biomechanical properties and cell compatibility [25]. The above shows the superior application prospect of PCL in medical materials.

Usnic acid (UA), also known as lichen acid, widely exists in lichen plants and is a natural furan antibiotic, which has been partially used in cosmetics, health products, and bacteriostatic agents [26,27]. Moreover, UA has good antibacterial effect and biological activity in vitro, and can effectively inhibit the growth of *E. coli*, *S. thermophilus*, *S. pyogenes,* and other bacteria [28]. It can be used to treat acne, seborrheic dermatitis, and other related skin diseases, and has a broad prospect of application in vitro [29,30]. Previous studies have shown that UA reduces bacterial infection in mouse models without detectable toxicity and drug resistance, which may bring bright prospects for UA in infection treatment [31,32,33]. Paula and colleagues prepared a composite membrane with the addition of usnic acid that improved wound healing in a mouse model [34].

Anthony Atala and his colleagues directly printed the fibrin gel containing rabbit chondrocytes on the surface of polycaprolactone fiber film prepared by electrostatic spraying using biological 3D printing. The research found that the mechanical strength of this composite material was more than four times higher than that of the single printed hydrogel material [35]. Liu et al. carried out multi-stage interactive printing by electrostatic spraying and bio-printing technology, and prepared multi-stage polycaprolactone micro fiber gel and drug-cartilage regeneration scaffold. The study found that the micro-nano structure had good biocompatibility and mechanical strength [36]. However, this method requires a layer of hydrogel material covered by 3D printing on the surface of the micro fiber material by electrostatic spraying and bio-printing technology, respectively, to form the composite structural material. As a result, the size and distribution of micro fibers are uneven, which greatly reduces the mechanical strength of the material and increases the production cost. If the electrostatic injection and bioprinting can be combined into one, the rational design and manufacture of electrostatic injection bioprinting equipment for the manufacture of microfiber structure gel composite materials is expected to develop a new type of microfiber structure electrospinning materials.

Therefore, in this work, biomaterials (PCL and collagen), active drugs (UA), and other components are directly prepared into microfiber structure biomaterials through near-field electrospinning (NFES) through reasonable design, which have good antibacterial performance, water absorption property, hydrophilicity, and biocompatibility, and are expected to be applied to skin wound treatment and repair, thus making up for the shortcomings caused by the recombination of materials with two different structures prepared by a single technology. Figure 1 is the flow chart of preparing PCL/collagen/UA composite fibers (PCUCF) by NFES.

## 2. Experimental Section

### 2.1. Materials Required

Polycaprolactone (PCL, Mn = 50,000 g/mol) was produced by the American Solvay company (Houston, TX, USA). Calf collagen type I (CO, Mn = 3000 g/mol) was bought from Xi’an Yunhe Biotechnology Co., Ltd., (Xi’an, China). Hexafluoro-2-propanol (HFIP) and usnic acid (UA) were purchased from Shanghai McLean Biochemical Technology Co., Ltd., (Shanghai, China). Agar was gained from Tianjin Fuchen Chemical Reagent Co., Ltd., (Tianjin, China). Beef extract and peptone bacteriological was acquired from Guangdong Huankai Microbiology Technology Co., Ltd., (Guangzhou, China). None of the above reagents had been purified further. Furthermore, the composition and shape of the main matrix materials have been indicated, and all other materials are pure reagents.

### 2.2. Experimental Methods

#### 2.2.1. Preparation and Electrospinning Process of PCL-Ordered Microfibers

The principle of NFES is shown in Figure 2 [37]. The needle head and the flat plate are connected by a high-voltage power supply. The air interface between the two will generate a very high local electric field, which is sufficient to overcome the interface surface tension, thus forming a Taylor cone and triggering a polymer jet. Before the polymer jet is dispersed, it is deposited directionally according to the specified trajectory to obtain ordered microfibers. The NFES machine was purchased from Foshan Qingzi Precision Measurement and Control Co., Ltd., China. The built-in precise control system can precisely control different parameters. The sample fibers were prepared along a square path into a grid dressing with 400 μm intervals.

In this work, the pure PCL polymer solution and single-layer PCL NFES fiber were prepared to determine the optimal process parameters. A total of 1.35 g PCL was added to 6.15 g HFIP, and the PCL was stirred magnetically for 2 h under 40 °C water bath heating until the PCL was completely dissolved to prepare 18% wt PCL/HFIP solution (recorded as solution I). The PCL/HFIP solution with different concentration gradients prepared by the same method was 16% wt and 20% wt, respectively. A disposable syringe with a specification of 10 mL was used to absorb about 5 mL of polymer solution, and was fixed to the syringe pump. The tip of the syringe with a model of 21 G was connected to the high-voltage generator and connected to the stainless steel receiving plate to prepare PCL ordered microfibers and optimize the process parameters of electrostatic spinning. The parameters were set as follows: the voltage was set as 1.7, 2.1, and 2.5 kV, the jet speed was adjusted between 0.2 and 0.6 mL/h, the platform moving speed was controlled between 120 and 180 mm/s, and the distance between the needle head and the stainless steel receiving plate was set as 1 mm, 2 mm, and 3 mm. The temperature and relative humidity were controlled at 35 °C and 50%, respectively, during the whole spinning process. After the electrospinning was completed, the fibers were taken down and freeze-dried for 12 h in a vacuum freeze dryer (FD-1A-50, Shanghai Hefan Instrument Co., Ltd., Shanghai, China) to remove excess solvent. PCL fibers obtained with different parameters were observed under a digital biological microscope (WSB12000, Guangzhou Microfield Optical Instrument Co., Ltd., Guangzhou, China), and then the average diameter was measured and calculated with Nano Measure software to determine the best near-field direct writing spinning parameters.

#### 2.2.2. Preparation of PCL/CO/UA-Ordered Composite Microfibers

NFES technology was used to organically combine PCL/CO/UA, and 20 layers of PCUCF (PCL/CO/UA Composite fiber) were prepared using the best near-field electrospinning parameters obtained above. First, the CO/HFIP mixed solution was previously prepared. A total of 0.15g CO was weighed and dissolved in 1.05g HFIP. After CO was completely dissolved by ultrasound for 2 min, it was added to solution I (keeping the mass ratio of PCL to CO at 8:2, which is recorded as P8C2). P9C1 and P7C3 were prepared in the same way. Then, 0.5%, 1%, and 1.5% UA of different mass concentrations were added to P8C2, and then magnetic stirring was used to mix them evenly to obtain a mixed solution of different gradient collagen and different gradient UA. Then, prepare PCUCF with the best NFES parameters in 2.3, and freeze dry the prepared PCUCF for 12h to remove any excess solvent.

#### 2.2.3. Simulation of Electrostatic Field in near Field Electrospinning

In order to theoretically explore the influence of near-field electrospinning parameters on fiber forming, the optimal voltage parameters obtained in Section 2.2.2 were calculated by using the electric field simulation module of COMSOL to calculate the high-voltage electric field potential formed after the spinning head tip was energized, and the distribution cloud diagram of potential and the distribution diagram of electric field line body arrow were obtained.

#### 2.2.4. Preparation of Collagen Crosslinking and Composite Electrospinning Fiber after Crosslinking

Genipin is far less toxic than glutaraldehyde and other commonly used chemical crosslinking agents [38]. A total of 5 g of collagen was dissolved in 50 g of genipin solution with different concentrations (0.1, 0.3, and 0.5 mmol/L), and cross-linked for 6 h at 25 °C. Then, the composite fiber with PCL/CO = 8:2 and 1% UA content was prepared by the method in Section 2.2.2 and recorded as AC-PCUCF (after crosslinking the PCL/CO/UA composite fiber).

### 2.3. Characterization

#### 2.3.1. FTIR Analysis

Collagen and KBr were mixed and ground at a ratio of 1:100 (*w/w*), dried at 50 °C and pressed, and tested with a Fourier transform infrared spectrometer (Nicolet IS10, Thermer Fisher Technology, Waltham, MA, USA). The scanning range was from 400 to 4000 cm^−1^, the scanning times were 32, and the resolution was 4 cm^−1^.

#### 2.3.2. SEM Analysis

A scanning electron microscope (EVO MA 15, Carl Zeiss Optics (China) Co., Ltd., Guangzhou, China) was used to observe the surface morphology of the samples. Before the test, the indication of gold-sprayed fiber with a thickness of 12 nm was used as the conductive layer using a vacuum coater (EM ACE600, Leica Instruments Co., Ltd., Wetzlar, Germany). The accelerating current of spraying gold was 30 mA, the gold content was 99.99%, and the sputtering distance was 6 cm.

#### 2.3.3. Mechanical Property Test

The tensile properties of composite fibers with different collagen concentrations (P9C1, P8C2, and P7C3) and PCUCF before and after crosslinking were tested by a universal mechanical testing machine (AGS-X, Shimadzu, Japan). To explore the effects of collagen concentration and crosslinking Genipin content on the mechanical properties of PCUCF, the sample was cut into a 1 × 4 cm rectangular sample and fixed on the fixture of the mechanical testing machine for tensile test. The sample thickness A was measured by the helical micrometer. In this process, the tensile speed of the universal testing machine was 50 mm/min. The tensile strength was calculated by Equation (1).
(1)$$P=\frac{F}{a\times b}$$
where P is tensile strength, F is the maximum tensile force the fiber mesh can withstand, a is mesh thickness, and b is mesh width.

The elongation at break can be calculated by Equation (2).
(2)$$\delta =\frac{L}{L-L_{0}}\;\times \;100\%$$
where δ represents elongation at break, L is the length of the fiber mesh at break, and L_0_ is the original length of the mesh.

#### 2.3.4. Antibacterial Performance Test

In this work, the beef extract peptone solid medium was prepared for culture observation. Four groups of PCUCF were sampled with 6 mm diameter discs using a hole punch and were sterilized for 4 hours under UV radiation. Then, the two bacteria were activated on a sterile clean workbench, and the activated bacteria were activated in a constant temperature and humidity box (LHS-150SC, Shanghai Yi Heng Technology Co., LTD., Shanghai, China) for 10 h. The activated E. coli and S. thermophilus were coated with beef peptone medium. Then, the prepared PCUCF was placed on the medium and sealed, and incubated in a constant temperature and humidity box at 37 °C for 12 h. The growth of bacteria and the formation of antibacterial zone around the experimental samples were observed and photographed. 

#### 2.3.5. Hydrophilic and Water Absorption Performance Test

Samples were cut into 2 × 4 cm flake, fixed on the slide, and then a high precision syringe was used to place a drop of deionized water on the surface (5 μL). After standing for 5 min, the contact angle was tested with the camera and image analysis system of the Optical Contact Angle Measuring Instrument (OCA20, Dataphysics, Filderstadt, Germany). Each group of samples was measured 5 times and the average value was taken.

The sample was cut into a square sample with a side length of 5 cm and weighed to record the original weight, denoted as m_0_. The sample was immersed in deionized water and placed for 24 h at the ambient temperature of 37 °C. It was then clamped out with tweezers. The weight of the sample after immersion was weighed and recorded as m. The moisture absorption rate can be calculated by Equation (3).
(3)$$\mathrm{Moisture}\;\mathrm{absorption}\;\mathrm{rate}\;=\frac{m\;-\;m_{0}}{m_{0}}\times 100\%$$

#### 2.3.6. Determination of Drug-Sustained Release Performance In Vitro

The solution with the concentration of 1, 2, 4, 8, and 10 μg/mL was prepared with PBS buffer. The absorbance of the solution was measured at the maximum absorption wavelength of 283 nm, and the standard absorption curve and fitting equation were obtained.

A complete 20 layers, 50 × 50 mm fibrous mat was taken and placed in 50 mL PBS buffer solution at a constant temperature of 37 °C. At different time points, 0.5 mL was sampled with a pipette gun, diluted 5 times with PBS buffer, and the absorption wavelength of the solution was measured at 283 nm. Then, substitute the standard curve to obtain the concentration of UA in the solution at this time, and then calculate the percentage of cumulative drug release at this time according to Equation (4).
(4)$$\mathrm{Percentage}\;\mathrm{of}\;\mathrm{cumulative}\;\mathrm{drug}\;\mathrm{release}\;=\frac{W_{t}}{W_{0}}\times 100\%$$
where W_t_ is the drug release amount at different time points, and W_0_ is the content of pine acid in the composite fiber.

#### 2.3.7. XRD Analysis

An X-ray diffractometer (D8ADVANCE, Bruker AXS, Karlsruhe, Germany) was used to evaluate the crystalline phases of the composite fibers. Measurements were made in the 2θ range of 5–80° at a scanning rate of 5 °/min using copper target radiation (λ = 0.154 nm) operating at 30 kV and 20 mA.

#### 2.3.8. Thermal Performance Analysis

The composite fiber was cut into powder, and the thermogravimetric analysis was carried out with a thermogravimetric analyzer (TG, NETZSCH Instrument Manufacturing Co., Ltd., Selb, Germany) from 25 °C to 600 °C at the rate of 10 °C/min to investigate the thermal denaturation temperature of the composite fiber. The thermal crystallization properties of the composite fibers were tested using a differential scanning calorimeter (DSC8000, Perking Elmer Co., LTD., Waltham, MA, USA) under the condition of a nitrogen flow rate of 50 mL/min in the range of 25–215 °C.

#### 2.3.9. Cell Proliferation Test In Vitro

Human umbilical vein endothelial cells (HUVEC) were used in the cell culture experiments to evaluate the proliferation of composite fibers in vitro. HUVEC cells were resuscitated, inoculated into a 25 mL cell culture flask, and cultured in a cell incubator (5% CO_2_, 37 °C) with a high-glucose medium containing 10% fetal bovine serum, 1% penicillin, and streptomycin double antibody. Cell growth was observed daily and the medium was changed every other day. The UV-sterilized materials were placed in 24-well cell culture plates with 4 parallel wells in each group. The previously cultured HUVEC was taken, cell counted, and diluted, and each well was inoculated with 600 μL cell suspension with a concentration of 1 × 10^7^/L in a 24-well cell culture plate containing the material for co-culture. Samples were co-cultured for 1, 3, 5, and 7 days at four time points. To each group, 60 µL CCK-8 reagent was added (medium: CCK-8 reagent 10:1) and incubated in a cell incubator at 37 °C for 2 h. After incubation, the culture medium was transferred to a 96-well plate cell culture plate, and the absorbance value (OD value) of each group in the 96-well plate was tested by an enzyme-labeled instrument (Multiskan GO, Thermo Scientific, Waltham, MA, USA) at the wavelength of 450 nm. Each sample was tested three times to obtain results.

## 3. Results and Discussion

### 3.1. Preparation and Electrospinning Process of PCL-Ordered Microfibers

Figure 3 shows the morphology of fibers with different electrospinning process parameters (in Table 1) observed under the microscope. As shown in Figure 3a,b1,b2, when the solution concentration was lower than 18% in the past, there were too many liquid drops on the fiber surface, which made the fiber appear obviously uneven, and the fiber morphology was poor (as shown in Figure 3b1). When the solution concentration is higher than 18%, the solution viscosity is too high, which easily leads to an unstable fiber diameter (as shown in Figure 3b2), and the needle is very easy to block, making the spinning process difficult. As shown in Figure 3a, the fiber diameter obtained from 18 wt% PCL solution is uniform (53.4 ± 1.6 μm). This may provide a preliminary basis for subsequent drug loading, release, and other functions. Therefore, the concentration of PCL was fixed at 18 wt% in this study.

As shown in Figure 3c1,c2, the fiber diameter decreases with the decrease of flow rate and increases with the increase of flow rate; similarly, as shown in Figure 3d1,d2, the fiber diameter increases with the decrease of the platform moving speed, and decreases with the increase of the platform moving speed. This is because the unit volume of fiber increases when the speed decreases under a certain flow rate, and vice versa. However, small fibers that are too small are not conducive to drug release. Fibers that are too large will lead to a large unit volume of fiber, slow solvent volatilization, and the same situation is true for b1, where the fiber surface is rough.

As shown in Figure 3e1,e2, the applied voltage can be spun in the range of 1.7–2.5 kV, and the fiber diameter decreases with the increase of voltage.

As shown in Figure 3f1,f2, when the distance between the tip and the platform is 1mm, the fiber surface appears obviously rough, which is caused by the short solvent evaporation time. When the distance between the tip and the platform is 3 mm, the fiber will twist. This is because the rear end of the Taylor cone, which has been formed when it is far away, has a similar trend to the traditional distal electrospinning.

It can be seen from the above results and analysis that the structure of the fiber and the diameter of the microfiber can be controlled by changing the process parameters of electrospinning, and the PCL concentration of 18% wt, the flow rate of 0.4 mL/h, the platform moving speed of 150 mm/s, the applied voltage of 2.1 kV, and the distance between the tip and the platform of 2 mm can be selected as the optimal process parameters for near-field electrospinning in this work.

### 3.2. Electrostatic Field Simulation of NFES

The analysis of potential distribution formed in the process of electrostatic spraying is shown in Figure 4, which is the potential distribution cloud diagram and the combination diagram of electric field line arrow distribution calculated with the optimal spinning voltage of 2.1kV. It can be seen in Figure 4 that the potential of the needle and its vicinity is the highest, the potential of the position farther away from the needle is the lower, and the potential of the receiving plate is the lowest. Moreover, the electric field lines are symmetrically distributed according to the spinning head, and the overall trend is that the spinning head shoots at the collecting plate. The electric field line on the central axis of the spinning needle vertically shoots at the collecting plate, which further verifies that the solution vertically shoots at the collecting plate from the spinning needle during actual printing.

### 3.3. FTIR Analysis

Figure 5a shows the FTIR spectra and corresponding wave number changes of collagen cross-linked by different amounts of genipin. On the whole, the collagen structure after crosslinking has not changed; in short, the basic structure of collagen will not be damaged by genipin crosslinking. In the infrared spectrum of collagen, there is a broad peak at 3297 cm^−1^, which is attributed to the stretching vibration of the O-H or N-H of collagen amide group, which is the characteristic adsorption of the amide A band. The peak value at 2958 cm^−1^ is designated as the characteristic amide B band of collagen, which is caused by the asymmetric stretching vibration of the C-H group of collagen. The peak value of 1639 cm^−1^ is attributed to the C=O stretching vibration of the amide I band. This is similar to the result of Feng et al [39]. As shown in Figure 5b, the amide I band of collagen crosslinked at different concentrations (0.1–0.5 mmol/L) moves towards high wave numbers. With the increase of genipin concentration, the movement of amide I band in collagen is more obvious, which means that genipin has a higher degree of crosslinking with collagen. Schiff base reaction argues that in the presence of crosslinking agent, the reaction can be indicated by the movement of the amide I band to a high wave number [40]. Therefore, the crosslinking reaction takes place through the Schiff base reaction between genipin and the amino group in collagen.

### 3.4. SEM Analysis

SEM observation of 20 layers of PCL fiber and PCUCF before and after crosslinking is shown in Figure 6, respectively. Highly ordered and layered composite spun fibers with perfectly arranged structures were successfully prepared. PCL fibers in Figure 6a have a smooth surface because it does not contain other substances. For PCUCF in Figure 7b, the fibrous surface appears to have obvious concavity and convexity, which may be caused by the slow solvent volatilization caused by the inhibition of the viscosity of collagen on the solvent volatilization after the addition of collagen. Then, looking at AC-PCUCF in Figure 6c, the cross-linked composite fiber collagen has better adhesion effect and more compact structure, producing a structure similar to “muscle”. In general, Figure 6a–c shows that the stacking of fibers is thin at the top and thick at the bottom. This is caused by the difference between near-field direct electrospinning and far-field electrospinning. The close distance between the needle head and the platform causes the solvent to volatilize insufficiently during the spinning process, resulting in downflow.

### 3.5. Mechanical Property Test

Figure 7a shows the mechanical property test results of composite fibers with different contents of collagen. The results showed that the higher the content of collagen was, the less tightly the PCL molecular chains were bound, and the lower the tensile strength and elongation at break of the fibers were. However, the content of 10% wt protein is an exception, because the addition of protein has bonded the fiber to a certain extent, which has slightly improved the elongation at break. As shown in Figure 7b, the tensile strength of the composite fiber with 20% content of collagen is only 7.6 ± 0.3 MPa. With the increase of genipin concentration, the tensile strength of the composite fiber continues to increase. When the concentration of genipin was 0.5 mmol/L, the tensile strength of the composite fiber reached 10.8 ± 0.9 MPa, and the elongation at break also increased slightly. It can be seen from this that the collagen chain of the composite fiber cross-linked by genipin becomes longer, leading to the tight connection of the spinning fibers, which leads to the enhancement of mechanical properties. This is consistent with the research results of Zhou et al. [41].

### 3.6. Antibacterial Performance Test

It can be found from Figure 8 that when composite fibers are inoculated on the bacterial surface, obvious bacteriostatic circles are observed in all plates, and the diameter of bacteriostatic circles is positively correlated with the UA concentration before the UA concentration reaches the threshold. The content of collagen has no effect on the antibacterial properties of the drug. UA showed good antibacterial ability whether it was gram-negative bacteria or gram-positive bacteria, and only 1% wt reached the antibacterial threshold. It is worth mentioning that, for all kinds of bacteria, the inhibition zone uniformly spreads around with PCUCF as the center, which indicates that the release of UA is continuous and uniform.

### 3.7. Hydrophilic and Water Absorption Performance Test

As shown in Figure 9, in the PCL/collagen fiber scaffold, with the increase of collagen content, the water contact angle of the fibers gradually decreased from 105 ± 2.3° to only 73 ± 1.5° for P7C3, and the fibers changed from super hydrophobic interface to strong hydrophilicity, indicating that the addition of collagen improved the hydrophilicity of the composite fibers. This is due to the existence of a large number of hydrophilic groups in collagen, which leads to the increase of its hydrophilicity. This is similar to Lee et al. It can be seen from Figure 10 that after the addition of collagen, the water absorption performance of P8C2 is significantly higher than that of PCL fiber and slightly higher than that of commercial gauze due to its good hydrophilicity. Figure 11 also shows that the water absorption performance of cross-linked AC-PCUCF is higher than that of PCUCF, because the cross-linked collagen makes the composite fiber have a more compact structure, which is more conducive to water absorption performance and retention. In general, the composite fiber has a strong water absorption performance (water absorption rate can reach 500%) whether or not the drug UA is added. It can be seen that the composite fiber can effectively absorb wound exudates as a medical dressing to avoid wound impregnation. To sum up, the addition of 20% collagen makes the composite fiber have better hydrophilicity and water absorption performance, and also has certain mechanical properties. Therefore, the following studies keep the collagen content at 20%.

### 3.8. Determination of Drug Sustained Release Performance In Vitro

The standard absorption curve equation of UA obtained through absorbance test and calculation is y = 0.01425x + 0.04686 (R^2^ = 0.9997). The results of the in vitro sustained drug release experiment are shown in Figure 11. It can be seen from the image that in the first 8 h of PCUCF, the release rate of UA is faster, and the release rate reaches 61%. The drug release rate is 84 ± 2% after 24 h. The rapid drug release in the first 8 h indicated that the sustained release effect of UA in this matrix is fast. After crosslinking, the release time of AC-PCUCF was significantly delayed, the UA is not completely released after 72h, and the release trend is still rising. This is because the crosslinking of genipin makes the structure of collagen more compact, which can make the slow release of drugs become sustained and slow, which can also be intuitively seen from the SEM image. In conclusion, the chemical structure of PCUCF can be changed by crosslinking to control the release behavior of UA, which brings convenience to the research of drug-sustained release systems.

### 3.9. XRD Analysis

It is obvious in Figure 12 that PCL fibers have three typical characteristic diffraction peaks of 21.3° (110), 21.9° (111), and 23.6° (200) [42]. Compared with pure PCL fibers, the characteristic diffraction peak of PCL in the fiber sample added with collagen is significantly improved, which indicates that the addition of collagen can promote the crystallization of PCL. In the composite fibers PCUCF and AC-PCUCF, there are no characteristic diffraction peaks of UA, indicating that UA exists in composite fibers in an amorphous form.

### 3.10. Thermal Performance Analysis

The TG curve of the fiber is shown in Figure 13a. At about 100 °C, the sample mass is slightly lost, because the residual water in the sample evaporates. At 220−370 °C, because of the thermal decomposition of UA, the sample has obvious mass loss. The great loss of sample quality at 370−440 °C is caused by the thermal degradation of collagen and PCL, which leaves ash after releasing gas. It can be seen from the DTG diagram that the maximum weight loss temperature of PCL fibers, PCUCF, and AC-PCUCF is about 410 °C. Compared with pure PCL and PCUCF, the addition of collagen hardly affects the thermal stability of the fibers. However, the weight loss rate of AC-PCUCF is much lower than that of PCUCF at the maximum weight loss temperature, and the final ash content is more. It can be seen that the cross-linked AC-PCUCF has better thermal stability than PCUCF. Wu and his colleagues have shown similar results in their research [43].

Figure 14 shows the DSC curve of the fibers, and the melting peak of PCL is around 60 °C. Because the collagen is amorphous, there is no melting peak of collagen after 60 °C. However, there is no characteristic melting peak of UA in the PCUCF before and after crosslinking, so UA is dispersed in the composite fibers in an amorphous form, which also verifies the structure of the previous XRD analysis. Compared with PCUCF and AC-PCUCF, it can be clearly seen that the crystallinity of AC-PCUCF is smaller than PCUCF. Therefore, the molecular orientation and crystallization of PCL can be changed through crosslinking.

### 3.11. Cell Proliferation Test In Vitro

As shown in Figure 15, CCK-8 analysis shows that the number of HUVEC cells on PCL and composite fibers increased significantly with the extension of culture time. Only PCL blank control and PCUCF showed differences on the first, third, and fifth days. Compared with P8C2 and blank control group pure PCL, the OD value of composite fibers P8C2 added with 20% collagen is significantly higher than that of pure PCL on the secenth day with the extension of time, indicating that the addition of collagen is more conducive to cell adhesion and proliferation. Moreover, the proliferation results of PCUCF and AC-PCUCF added with UA are similar to P8C2, which also show good cell compatibility. These results indicate that HUVEC cells have good adhesion and proliferation ability on composite fibers P8C2, PCUCF, and AC-PCUCF. More importantly, the composite fibers has good biocompatibility for cell growth and proliferation in vitro.

## 4. Conclusions

In this work, NFES was used to prepare a new type of PCL/collagen/UA composite fibers with perfect arrangement and distinct layers. The optimum spinning concentration of PCL in solvent HFIP was determined to be 18%. When PCL/collagen = 8:2, it showed excellent hydrophilicity and water absorption performance. The UA content of 1% wt (calculated by the total mass of PCL and collagen) is the addition amount of composite fibers to achieve the best antibacterial performance. The microfibers have good antibacterial, hydrophilic, and water absorption performance. In addition, the cross-linked composite fibers have stronger mechanical properties, water absorption performance, and thermal properties. More importantly, the composite fibers can promote cell adhesion and proliferation and have good biocompatibility. In a word, PCL/collagen/UA composite fibers have potential applications in biomedical materials.

## Figures and Tables

**Figure 1 polymers-15-00223-f001:**
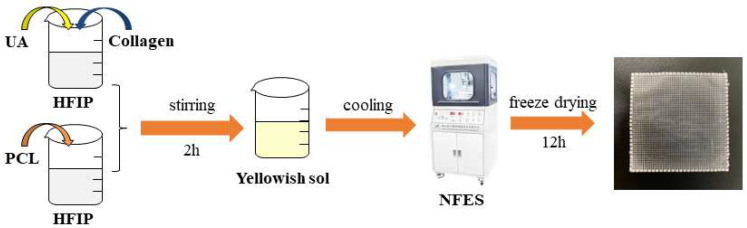
Flow chart of PCUCF prepared by NFES.

**Figure 2 polymers-15-00223-f002:**
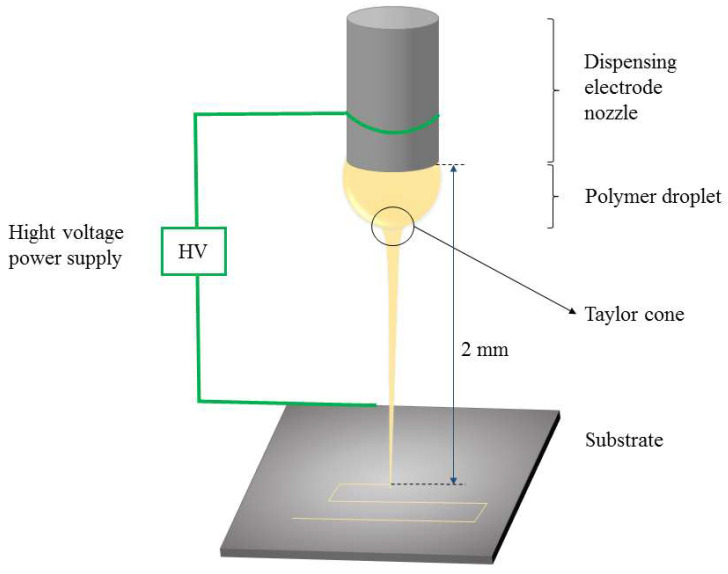
Schematic Diagram of NFES.

**Figure 3 polymers-15-00223-f003:**
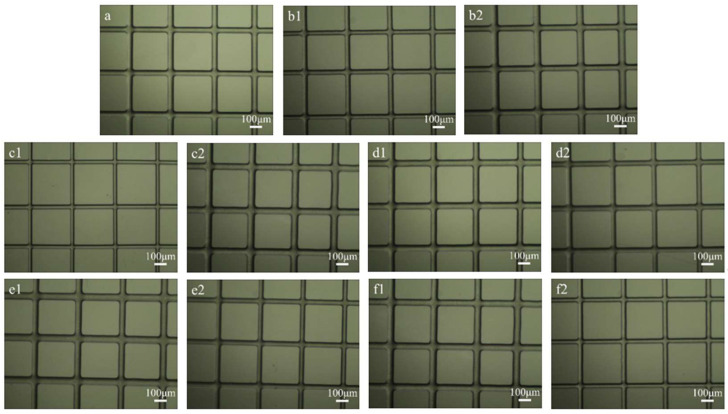
NFES of PCL with different process parameters. Wherein (**a**) is the microscope photo of PCL single-layer spinning fiber under the parameters of 18% wt concentration, flow rate 0.4 mL/h, voltage 2.1 kV, distance from tip to platform 2 mm, and platform moving speed 150 mm/s; (**b1**,**b2**): different concentrations; (**c1**,**c2**): different flows; (**d1**,**d2**): different moving speeds of the platform; (**e1**,**e2**): different voltages; (**f1**,**f2**): different distances from the needle tip to the platform.

**Figure 4 polymers-15-00223-f004:**
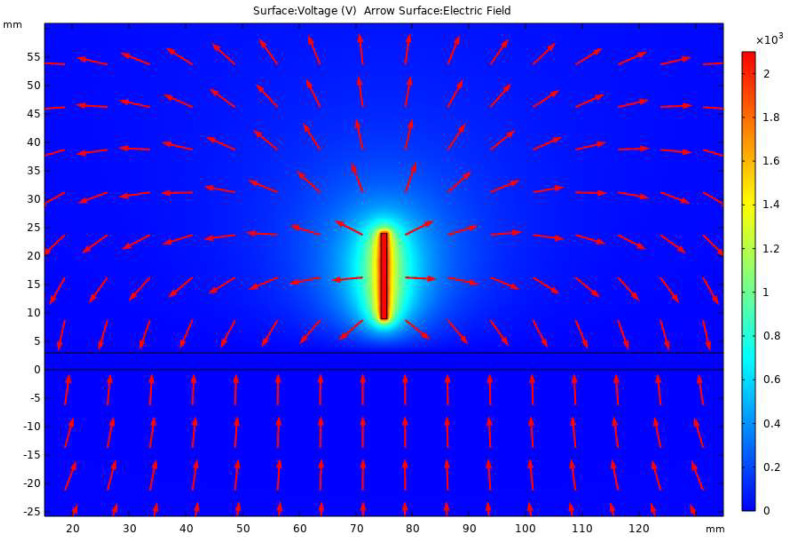
Distribution of electric field line arrows.

**Figure 5 polymers-15-00223-f005:**
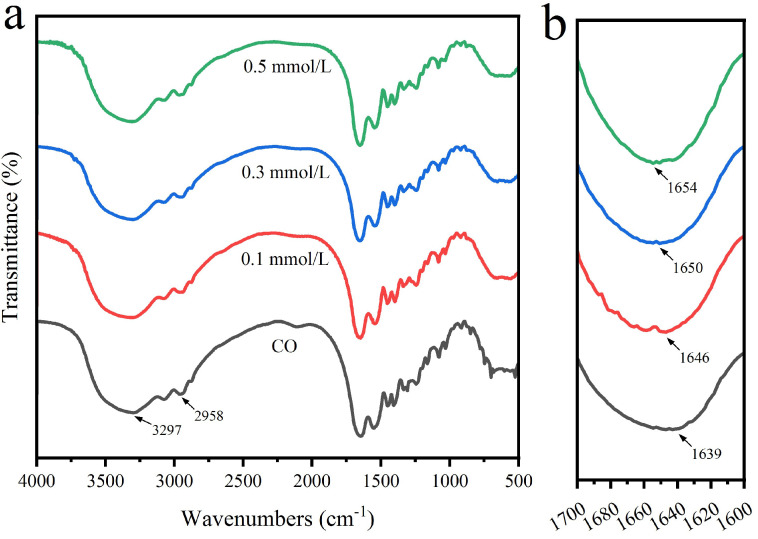
Infrared spectrum of collagen cross-linked with genipin at different concentrations: (**a**) full spectrum; (**b**) partial magnification at wave number 1600−1700 cm^−1^.

**Figure 6 polymers-15-00223-f006:**
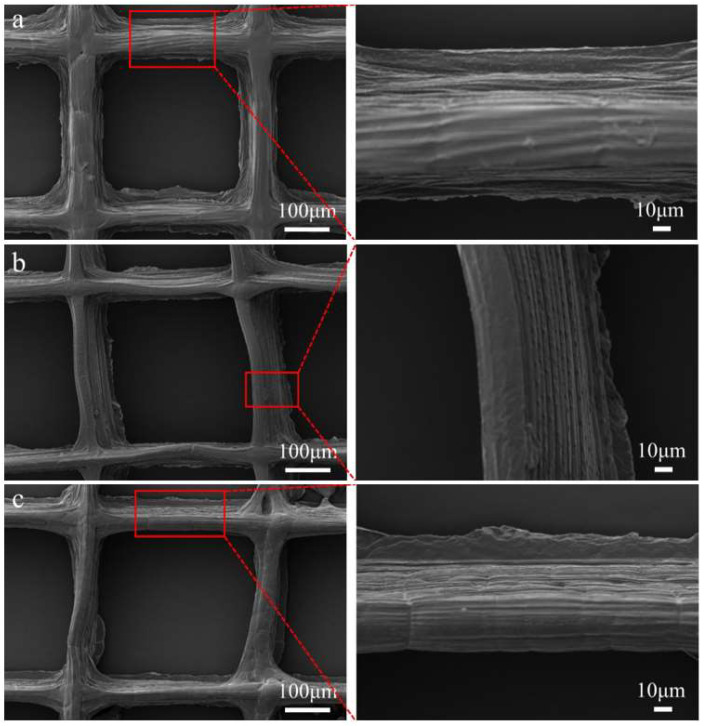
SEM diagram of fiber: (**a**) PCL; (**b**) PCUCF; (**c**) AC-PCUCF.

**Figure 7 polymers-15-00223-f007:**
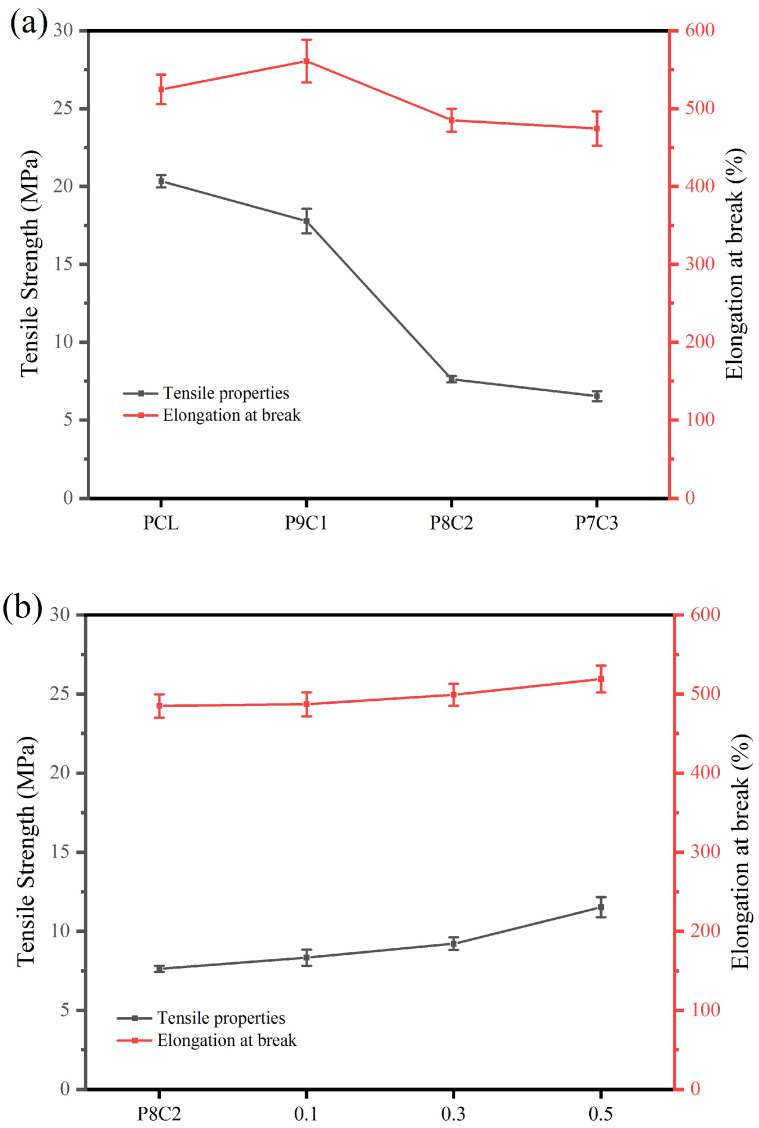
Schematic diagram of tensile and flexural properties of composite fiber: (**a**) different content of collagen; (**b**) crosslinking of genipin at different concentrations (0.1, 0.3 and 0.5 mmol/L).

**Figure 8 polymers-15-00223-f008:**
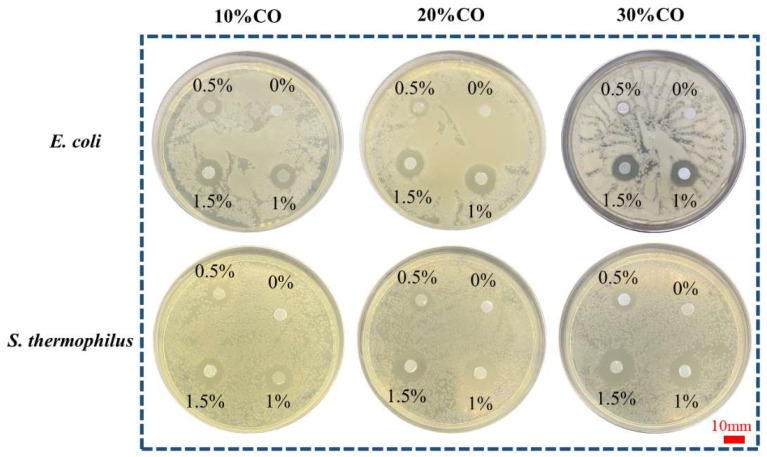
Effect of composite fibers with different content of collagen and different UA content (the number in the culture dish is the content) on the antibacterial properties of *E. coli* and *S. thermophilus*.

**Figure 9 polymers-15-00223-f009:**
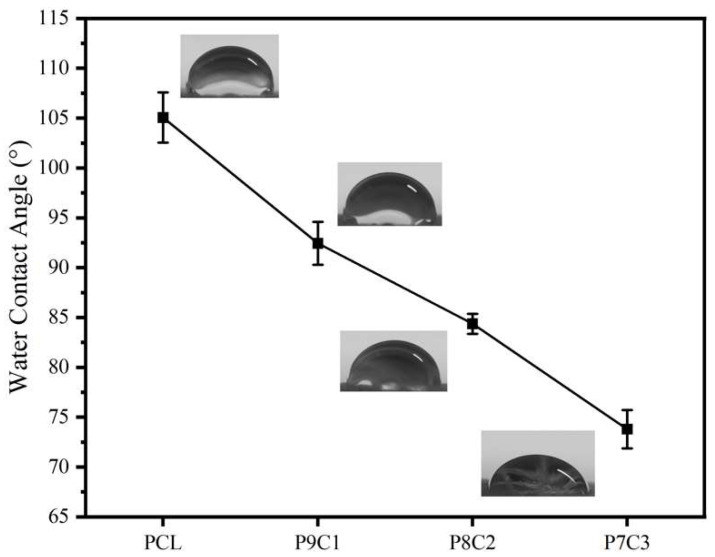
Schematic diagram of water contact angle of fibers.

**Figure 10 polymers-15-00223-f010:**
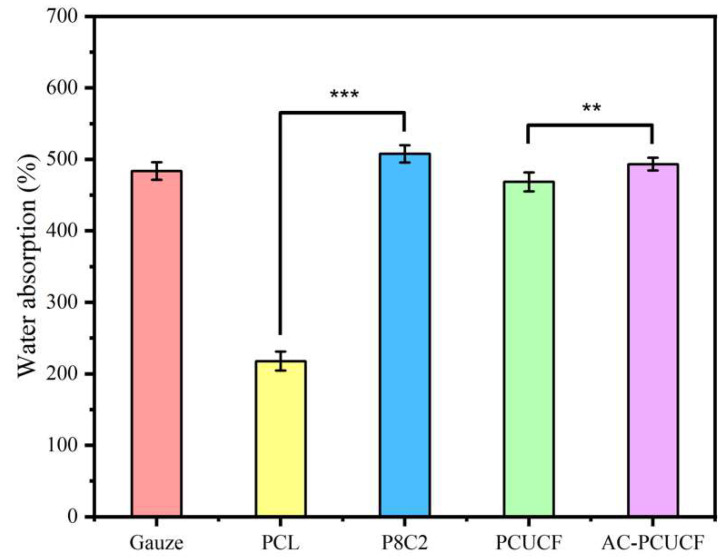
Schematic diagram of water absorption performance of fiber (* *p* ≤0.05; ** *p* ≤ 0.01; *** *p* ≤ 0.001; the same below).

**Figure 11 polymers-15-00223-f011:**
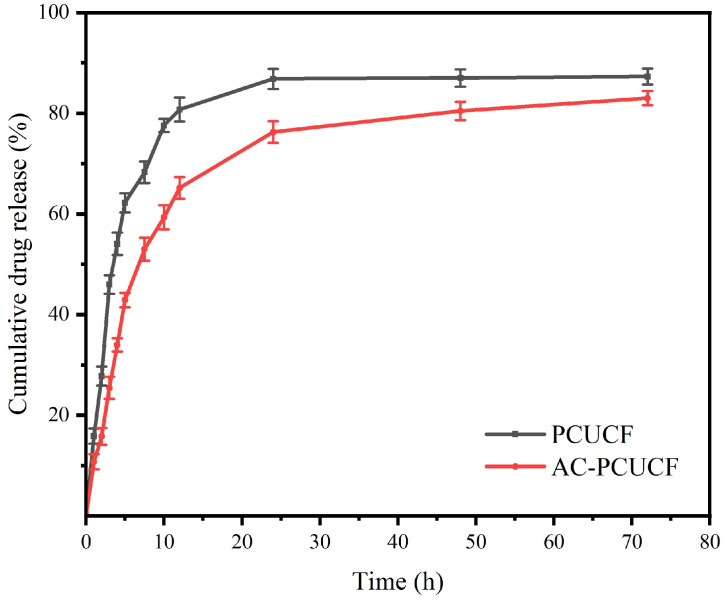
Drug release curve of composite fibers.

**Figure 12 polymers-15-00223-f012:**
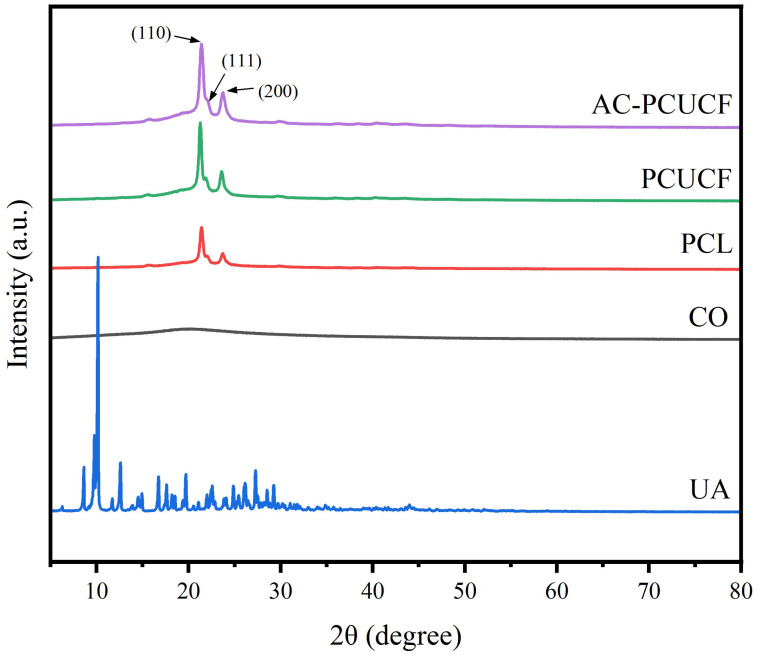
XRD analysis of composite fibers.

**Figure 13 polymers-15-00223-f013:**
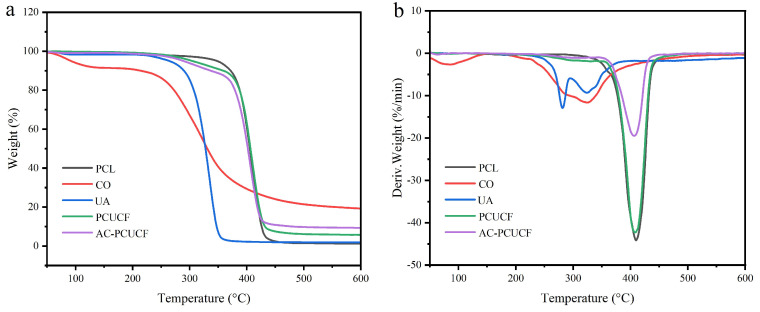
TG (**a**) and DTG (**b**) analysis of fibers.

**Figure 14 polymers-15-00223-f014:**
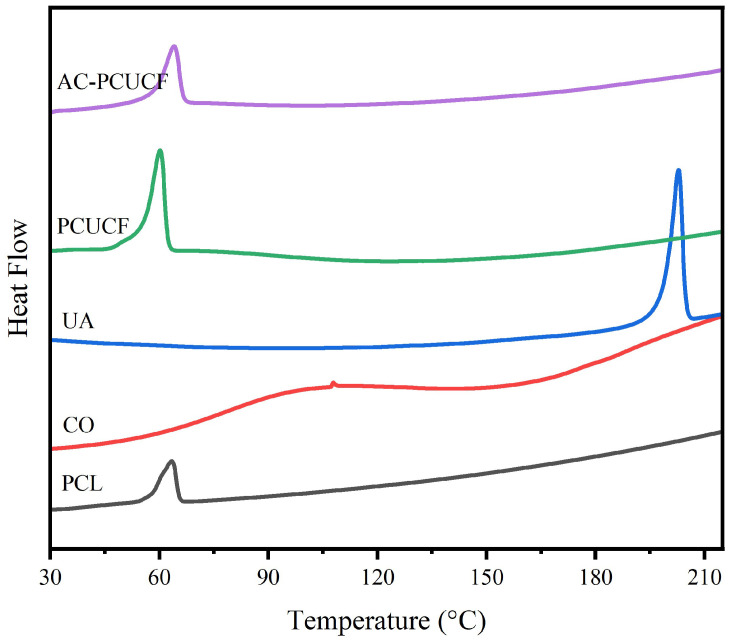
DSC analysis of fibers.

**Figure 15 polymers-15-00223-f015:**
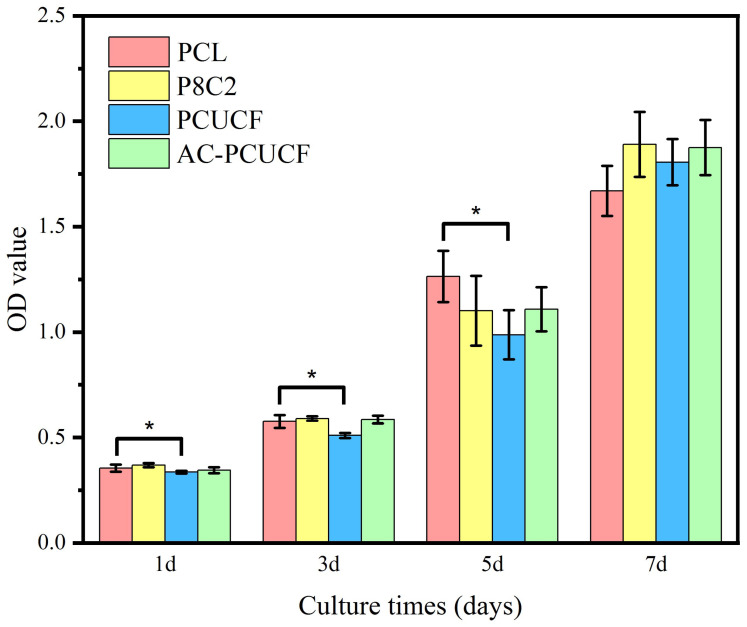
Cell behavior on composite fibers.

**Table 1 polymers-15-00223-t001:** Process parameter setting and fiber diameter distribution during electrospinning.

Samples (Figures)	wt (%)	Flow Rate (mL/h)	Applied Voltage (kV)	Distance (mm)	Moving Speed (mm/s)	Diameter (μm)
a	18	0.4	2.1	2	150	53.4 ± 1.6
b1	16	0.4	2.1	2	150	50.1 ± 1.6
b2	20	0.4	2.1	2	150	64.7 ± 5.5
c1	18	0.2	2.1	2	150	34.8 ± 2.2
c2	18	0.6	2.1	2	150	63.9 ± 3.9
d1	18	0.4	2.1	2	120	61.9 ± 1.6
d2	18	0.4	2.1	2	180	51.4 ± 1.5
e1	18	0.4	1.7	2	150	63.2 ± 2.2
e2	18	0.4	2.5	2	150	48.8 ± 1.4
f1	18	0.4	2.1	1	150	68.5 ± 2.0
f2	18	0.4	2.1	3	150	46.2 ± 2.4

## Data Availability

Data sharing not applicable.

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
