# Peer review of "PCL/Collagen/UA Composite Biomedical Dressing with Ordered Microfiberous Structure Fabricated by a 3D Near-Field Electrospinning Process"

_polymers, 2022, doi:10.3390/polym15010223_

Round 1

Author Response

Dear reviewer,                                                   

Thank you very much for your kind attention and careful review for our manuscript entitled “PCL/Collagen/UA composite biomedical dressing with ordered microfiberous structure fabricated by a 3D near-field electrospinning process”. We have carefully checked and revised our manuscript according to the suggestions of the reviewers. The responds were shown as follows:

  1. The phrase at the beginning of the abstract “Currently, electrospinning technology has made significant progress in the preparation and application of biomedical materials, but antibacterial biomedical materials with excellent antibacterial performance and high biological description are still hot and difficult in this field. Near field electrospinning (NFES) combines biological 3D printing and traditional electrospinning technology, which has unique advantages in drug loading and complex structure spinning" is more relevant to the introduction, and should be moved to the "Introduction" section.

Responds: Thank you very much for your kind suggestion. In the revised manuscript, we have reformulated the abstract as follows,

 “In this work, a functionalized polycaprolactone (PCL) composite fiber combining calf type I collagen (CO) and natural drug usnic acid (UA) was prepared, in which UA was used as an antibacterial agent. Through 3D near-field electrospinning, the mixed solution was prepared into PCL/CO/UA composite fibers (PCUCF), which has a well-defined perfect arrangement structure. The influence of electrospinning process parameters on fiber diameter was investigated, and the optimal electrospinning parameters were determined, and the electric field simulation was conducted to verify the optimal parameters. The addition of 20% collagen made the composite fiber have good hydrophilicity and water absorption property. In the presence of PCUCF, 1% UA content significantly inhibited the growth rate of gram-positive and negative bacteria in plate culture. The AC-PCUCF prepared by crosslinking collagen with genipin showed stronger mechanical properties, water absorption property, thermal stability and drug release perfor-mance. Cell proliferation experiments showed that PCUCF and AC-PCUCF had no cytotoxicity and could promote cell proliferation and adhesion. The results show that PCL/CO/UA composite fiber has potential application prospects in biomedical dressing.”

  1. In the introduction, it is necessary to formulate the purpose and objectives of the study.

Responds: Thank you very much for your kind suggestion. In the revised manuscript, the last paragraph of the introduction has elaborated the content to be studied and the expected results. Thanks again.

  1. In part 2.2.4, the temperature at which the collagen was cross-linked with heparin should be indicated.

Responds: Thank you very much for your kind suggestion. The crosslinking temperature is 25 ℃, which has been added to 2.2.4. 

  1. Comments on Fig.3 are not obvious when considering Fig.3. It may be the same as in the case of Fig. 7 to make explanations directly in fig. 3.

Responds: Thank you very much for your kind suggestion. Fig.3 shows the process investigation of electrospun single-layer fiber, and fig.7 shows the electron microscope picture of 20 layers of fiber. The two pictures are to illustrate different problems, so they are explained separately.

  1. In fig. 12 you need to remove two zeros at the origin. It is incorrect to indicate two zeros in the figure.

Responds: Thank you very much for your kind suggestions. In the revised manuscript, fig.12 has removed an origin. 

  1. In the list of references, more than a third of the references refer to publications before 2017. You can replace some of the references with later ones, since the topic of the article is very relevant, there have been a lot of works on this topic in the last few years.

Responds: Thank you very much for your kind suggestions. In the revised manuscript, most of the references before 2017 have been updated to those of recent years.

Reviewer 2 Report

·          Line 142: high pressure should by high voltage

·          Line 148: The NFES process needs more explanation. What is the size and the material of the substrate? How is the planned moving path, distance between lines? Is it a bought NFES setup or self-constructed? How is the height above the substrate measure/regulated?

·          Line 172: “optimal voltage parameters” Which condition is searched for to get the optimal voltage?

·          Line 175: word “nephogram” is unknown

·          Line 179: “genipin can be cross-linked with proteins” If genipin is a crosslinking material then change it to

·          “genipin can cross-link proteins …”. Genipin crosslinks collagen, not the other way around.

·          Line 184: So it is possible to do NFES after crosslinking?

·          There are two names of the crosslinking agent “genipin” in line 179 and “Gennipine” in line 203.

·          Line 228: “injected in the surface of a drop of deionized water” change to “place a drop of deionized water on the surface.”

·          Line 235: What is the procedure after taking the sample out of the water? Is it dried a bit?

·          Line 242: “fiber” Is fibrous mat meant? One fiber can not be 50x50mm.

·          Line 254: wavelength of Cu has no unit, probably nm. What is the angular step size?

·         Line 295 and Line 314/315: “fiber appear obviously uneven”, “fiber surface appears obviously rough, which is caused by the short solvent evaporation time“ and “the fiber will twist” Have the correct images been used? I don’t see twist, rough and uneven. If the evaporation time is short, I would expect to see which line has been printed first. The crossing points of the fibers have no height difference, so it seems the fibers have still been liquid when the second fiber was printed on the top. This could be added.

·         Fig. 4 shows a lot of unnecessary space. It is suggested to zoom in and show just one end of the needle and the collector, like x from 50mm to 100mm and y from 0mm to 10mm. Fig. 4 can be removed as the voltage is also shown in Fig. 5.

·         Line354: “agures” is not known

Author Response

Dear reviewer,                                                   

Thank you very much for your kind attention and careful review for our manuscript entitled “PCL/Collagen/UA composite biomedical dressing with ordered microfiberous structure fabricated by a 3D near-field electrospinning process”. We have carefully checked and revised our manuscript according to the suggestions of the reviewers. The responds were shown as follows:

  1. Line 142: high pressure should by high voltage.

Responds: Thank you very much for your kind suggestion. In the revised manuscript, the high pressure in line 142 has been changed to high voltage.

  1. Line 148: The NFES process needs more explanation. What is the size and the material of the substrate? How is the planned moving path, distance between lines? Is it a bought NFES setup or self-constructed? How is the height above the substrate measure/regulated?

Responds: Thank you very much for your kind suggestion. In the revised manuscript, more explanations of the NFES process have been added as follows,

  “NFES machine was purchased from Foshan Qingzi Precision Measurement and Control Co., Ltd., China. The built-in precise control system can precisely control different param-eters. The sample fibers were prepared along a square path into a grid dressing with 400 μm intervals.”

  1. Line 172: “optimal voltage parameters” Which condition is searched for to get the

optimal voltage?

Responds: Thank you very much for your kind suggestion. The optimal voltage is 2.1V, which is explained in 3.1 in detail.

  1. Line 175: word “nephogram” is unknown.

Responds: Thank you very much for your kind suggestion. “nephogram” is cloud diagram, in the revised manuscript, we have corrected it.

  1. Line 179: “genipin can be cross-linked with proteins” If genipin is a crosslinking material then change it to “genipin can cross-link proteins …”. Genipin crosslinks collagen, not the other way around.

Responds: Thank you very much for your kind suggestion. In the revised manuscript, more explanations of the NFES process have been added as follows,

  “As an excellent natural biological cross-linking agent, genipin can cross-link proteins, collagen, gelatin and chitosan to make biomaterials.”

  1. Line 184: So it is possible to do NFES after crosslinking? There are two names of the crosslinking agent “genipin” in line 179 and “Gennipine” in line 203.

Responds: Thank you very much for your kind suggestions. NFES can be carried out after crosslinking, which is recorded as AC-PCUCF, as mentioned below. The full text should be genipin, and we have corrected it in the revised manuscript.

  1. Line 228: “injected in the surface of a drop of deionized water” change to “place a drop of deionized water on the surface.”

Responds: Thank you very much for your kind suggestion. In the revised manuscript, we have corrected it.

  1. Line 235: What is the procedure after taking the sample out of the water? Is it dried a bit?

Responds: Thank you very much for your kind suggestion. It is measured with tweezers under the condition of keeping wet.

  1. 9. Line 242: “fiber” Is fibrous mat meant? One fiber can not be 50x50mm.

Responds: Thank you very much for your kind suggestion. Yes, “fiber” is fibrous mat, in the revised manuscript, we have corrected it.

  1. 10. Line 254: wavelength of Cu has no unit, probably nm. What is the angular step size?

Responds: Thank you very much for your kind suggestion. The radiation wavelength of the copper target is 0.154 nm, and the angle is 5 to 80 degrees, which has been added to the revised manuscript.

  1. 11. Line 295 and Line 314/315: “fiber appear obviously uneven”, “fiber surface appears obviously rough, which is caused by the short solvent evaporation time” and “the fiber will twist” Have the correct images been used? I don’t see twist, rough and uneven. If the evaporation time is short, I would expect to see which line has been printed first. The crossing points of the fibers have no height difference, so it seems the fibers have still been liquid when the second fiber was printed on the top. This could be added.

Responds: Thank you very much for your kind suggestion. The correct image has been used. When you zoom in on Figures b1 and b2, you can clearly see that the fibers are not straight but wavy. In Figure f1, the closer the distance is, the slower the solvent volatilizes, and the fiber surface becomes rough without volatilization, which can be clearly seen in the enlarged picture.

  1. 12. Fig. 4 shows a lot of unnecessary space. It is suggested to zoom in and show just one end of the needle and the collector, like x from 50mm to 100mm and y from 0mm to 10mm. Fig. 4 can be removed as the voltage is also shown in Fig. 5.

Responds: Thank you very much for your kind suggestion. Figure 5 does contain the content of Figure 4. In the revised manuscript, we have deleted Figure 4.

  1. 13. Line354: “agures” is not known.

Responds: Thank you very much for your kind suggestion. "Graces" is our mistake, which should be "argues", and has been corrected in the revised manuscript.
